



# The Impacts of Biomass Burning Activities on Convective Systems in the Maritime Continent

Hsiang-He Lee[1][*][@] and Chien Wang[1,2][**]

[1] Center for Environmental Sensing and Modeling, Singapore-MIT Alliance for Research and Technology, Singapore

[2] Center for Global Change Science, Massachusetts Institute of Technology, Cambridge, MA, U.S.A.

[*]Now at Atmospheric, Earth, and Energy Division, Lawrence Livermore National Laboratory, Livermore, CA, U.S.A.

[**]Now at Laboratoire d'Aerologie/CNRS/University of Toulouse, Toulouse, France

[@]Corresponding author address: Dr. Hsiang-He Lee, 7000 East Avenue, Livermore, CA, 94550, U.S.A.

E-mail: lee.hsianghe@gmail.com



## 34   Abstract

Convective precipitation associated with Sumatra squall lines and diurnal rainfall
over Borneo is an important weather feature of Maritime Continent in Southeast Asia.
Over the past few decades, biomass burning activities have been widespread during
summertime over this region, producing massive fire aerosols.  These additional aerosols
brought to the atmosphere, besides influencing local radiation budget through directly
scattering and absorbing sunlight, can also act as cloud condensation nuclei or ice nuclei
to alter convective clouds and precipitation in the Maritime Continent via the so-called
aerosol indirect effects.  Based on four-month simulations with or without biomass
burning aerosols conducted using the Weather Research and Forecasting model with
chemistry package (WRF-Chem), we have investigated the aerosol-cloud interactions
associated with the biomass burning aerosols in the Maritime Continent.  Results from
selected cases of convective events have shown significant impacts of fire aerosols on the
weak convections in hydrometeors and rainfall amount either in the Sumatra or Borneo
region.  Statistical analysis over the fire season also suggests that fire aerosols have
substantial impacts on the nocturnal convections associated with the local anticyclonic
circulation in the western Borneo.  In addition, near surface heating from absorbing
aerosols emitted from fires could weaken land breezes and thus the convergence of
anticyclonic circulation.  Therefore, the rainfall intensity of the nocturnal convections has
been significantly decreased during the fire events.




# 1   Introduction


Biomass burning in Southeast Asia has become a serious environmental and societal
issue in the past decade due to its impact on local economy, air quality, and public health
(Miettinen et al., 2011; Kunii et al., 2002; Frankenberg et al., 2005; Crippa et al., 2016;
Lee et al., 2018).   Abundant aerosols emitted from such fires not only cause
environmental issues but also affect regional weather and climate through the direct and
indirect effects of biomass burning aerosols (Grandey et al., 2016; Hodzic and Duvel,
2017; Jeong and Wang, 2010; Ramanathan and Carmichael, 2008; Taylor, 2010; Tosca et
al., 2013).   Carbonaceous compounds such as black carbon (BC) in biomass burning
aerosols can reduce sunlight through both absorption and scattering to warm the
atmosphere while cool the Earth's surface (Fujii et al., 2014; Andreae and Gelencsér,
2006; Satheesh and Ramanathan, 2000; Ramanathan et al., 2001).   Besides these direct
effects, biomass burning aerosols can act as cloud condensation nuclei or ice nuclei to
alter cloud microphysical structures and thus cloud radiation.  Such "indirect effects" of
these aerosols on the climate are even more complicated due to various cloud and
meteorological conditions (Sekiguchi et al., 2003; Lin et al., 2013; Wu et al., 2013;
Grandey et al., 2016; Ramanathan et al., 2001; Wang, 2004).
For the Maritime Continent in Southeast Asia, convective precipitation associated
with the so-called Sumatra squall lines (SSL) and diurnal rainfall over Borneo is an
important weather feature (Lo and Orton, 2016; Ichikawa and Yasunari, 2006; Koh and
Teo, 2009; Yi and Lim, 2006; Wu et al., 2009).  Convections of SSL are initially formed
in the northwestern side of Sumatra by the prevailing sea breezes from Indian Ocean and
the Sumatran mountain range, then propagate over the Malacca Strait affecting the Malay



Peninsula. Lo and Orton (2016) analyzed 22-year (1988 to 2009) ground-based Doppler
radar data and identified a total of 1337 squall lines in Singapore. They found that these
events with the diurnal cycle of rainfall most occur during either the summer monsoon
season (June-September) or the inter-monsoon periods (April-May and October-
November). Singapore, for example, experiences typically about 6~7 squall lines per
month during these periods. Oki and Musiake (1994) analyzed the seasonal and diurnal
cycles of precipitation using rain gauge data and showed that large-scale low-level winds
are a critical modulating factor in the diurnal cycle the convective rainfall over Borneo
besides the general reason of land-sea contrast behind convective rainfall in the Maritime
Continent. Furthermore, Ichikawa and Yasunari (2006) used five years Tropical Rainfall
Measuring Mission (TRMM) precipitation radar (PR) data to investigate the role of the
low-level prevailing wind in modulating the diurnal cycle of rainfall over Borneo. They
found that the diurnal cycle is associated with intraseasonal variability in the large-scale
circulation pattern, with regimes associated with either low-level easterlies or westerlies
over the island.
Interestingly, frequent biomass burning activities coincide with vigorous convective
systems in the Maritime Continent, especially during the summer monsoon season (June-
September), and could thus produce aerosols to affect convections in the region.
Rosenfeld (1999) analyzed TRMM data and hypothesized that abundant biomass burning
aerosols could practically shut off warm rain processes in tropical convective clouds.
Compared to the adjacent tropical clouds in the cleaner air, clouds encountered with
smokes could grow to higher altitudes with rain suppressed, hypothetically due to the
reduction of coalescence efficiency of smaller cloud drops into raindrops. Recently,





using Weather Research and Forecasting model with Chemistry (WRF-Chem), Ge et al.
(2014) have studied the direct and semi-direct radiative effects of biomass burning
aerosols over the Maritime Continent and found the radiative effect of biomass burning
aerosols could alter planetary boundary layer (PBL) height, local winds (including sea
breeze), and cloud cover. However, relative coarse resolution (27 km) adopted in their
simulation would not be able to reveal more details about how biomass burning aerosols
affect convective clouds through modifying cloud microphysics processes. Whereas,
Hodzic and Duvel (2017) have conducted a 40-day simulation using WRF-Chem with a
convection-permitting scale (4 km) to study the fire aerosol-convection interaction during
boreal summer in 2009 near the central Borneo mountainous region. Their result
suggests that modifications of the cloud microphysics by biomass burning aerosols could
reduce shallow precipitation in the afternoon and lead to a warm PBL anomaly at sunset,
all lead to an enforcement of deep convection at night. However, they have also
indicated that the radiative processes of moderately absorbing aerosols tend to reduce
deep convection over most regions due to local surface cooling and atmosphere warming
that increase the static stability, hence suggesting the complexity of the interaction of
biomass burning aerosols and convective clouds in the Maritime Continent.

In this study, we aim to examine and quantify the impacts of biomass burning

aerosols on convective systems over two targeted regions for analyses: the northern
Sumatra and the western Borneo in the Maritime Continent. Our focus is on not only the
change of hydrometeors in the convective clouds but also the change of rainfall amount
and intensity in these regions. We firstly describe methodologies adopted in the study,
followed by the results and findings from our numerical simulations over the Maritime



Continent. We have selected three cases in each study region to perform detail analyses.
In addition, statistical analyses covering the entire modeled fire season for each of these
two regions have also been performed to provide more generalized pictures about the
effects of fire aerosol on convection. The last section summarizes and concludes our
work.

## 2  Methodology

### 2.1  Model and emission inventories

In order to simulate trace gases and particulates interactively with the meteorological
fields, the Weather Research and Forecasting model coupled with a chemistry module
(WRF-Chem, see Grell et al. (2005)) version 3.6.1 is used in this study. Within WRF-
Chem, the Regional Acid Deposition Model, version 2 (RADM2) photochemical
mechanism (Stockwell et al., 1997) coupled with the Modal Aerosol Dynamics Model for
Europe (MADE) as well as the Secondary Organic Aerosol Model (SORGAM)
(Ackermann et al., 1998; Schell et al., 2001) are included to simulate atmospheric
chemistry and anthropogenic aerosol evolutions. MADE/SORGAM uses a modal
approach to represent the aerosol size distribution and predicts mass and number
concentrations of three aerosol modes (Aiken, accumulation, and coarse).
To resolve the convective system in the Maritime Continent in our simulations, two
model domains with two-way nesting are designed. Here, Domain 1 (431 × 141 grid
cells) has a resolution of 25 km, while Domain 2 (561 × 591 grid cells) has a resolution
of 5 km (Fig. 1). Specifically, Domain 1 is positioned to include the tropical Indian
Ocean on its west half in order to capture the path of Madden-Julian Oscillation (MJO),



and in the meantime to have a northern boundary constrained within 23°N in latitude to
avoid potential numerical instability from the terrain of Tibetan Plateau. Domain 2 with
a finer resolution is positioned to cover the mainland Southeast Asia as well as the islands
of Sumatra and Borneo. The National Center for Environment Prediction FiNaL (NCEP-
FNL) reanalysis data (National Centers for Environmental Prediction, 2000) are used to
provide initial and boundary meteorological conditions, and to perform four-dimensional
data assimilation (FDDA) to nudge model temperature, water vapor, and zonal and
meridional wind speeds above the planetary boundary layer (PBL) for Domain 1. The
Mellor-Yamada-Nakanishi-Niino level 2.5 (MYNN) (Nakanishi and Niino, 2009) is
chosen as the scheme for planetary boundary layer in this study. Other physics schemes
adopted in the simulations include Morrison two-moment microphysics scheme
(Morrison et al., 2009), RRTMG longwave and shortwave radiation schemes (Mlawer et
al., 1997; Iacono et al., 2008), Unified Noah land-surface scheme (Tewari et al., 2004),
and Grell-Freitas ensemble cumulus scheme (Grell and Freitas, 2014) (for Domain 1
only).
WRF-Chem needs emissions for gaseous and particulate precursors to drive its
simulations. For this purpose, we have used the Regional Emission inventory in ASia
(REAS) version 2.1 (Kurokawa et al., 2013). REAS includes emissions of most primary
air pollutants and greenhouse gases, covering each month from 2000 to 2008. In
addition, the Fire INventory from U.S. National Center for Atmospheric Research
(NCAR) version 1.5 (FINNv1.5) (Wiedinmyer et al., 2011) is also used in the study to
provide biomass burning emissions. FINNv1.5 classifies burnings of extratropical forest,
tropical forest (including peatland), savanna, and grassland. Fire heat fluxes for four



different types of fire are prescribed in WRF-Chem to calculate the plume height (rf.
Table 1 in Freitas et al. (2007). For peatland fire, we have set its heat flux as 4.4 kW m$^{-2}$,
which is the same as that of savanna burning and differs from that of the tropical forest
burning in 30 kW m$^{-2}$. The modified the plume rise algorithm in WRF-Chem to
specifically improve the representation of tropical peat fire has been described in Lee et
al. (2017). It is worth indicating that the heat flux from biomass burning is not
incorporated in thermodynamic equation of current WRF-Chem model.

The default chemical profiles of several species in the lateral boundary condition are

higher than their background concentrations in our study region and thus equivalent to
provide additional aerosol sources from boundaries. To prevent this, we have set NO,
NO$_2$, SO$_2$, and all primary aerosol levels to zero at the lateral boundaries of Domain 1.
We have also adjusted the ozone profile used for lateral boundary condition based on the
World Meteorological Organization (WMO) Global Atmosphere Watch (GAW) station
in Bukit Kototabang, Indonesia (Lee et al. (2019).

## 184  2.2  Numerical experiment design

Two numerical simulations, both include fossil fuel emissions while either with and

without the biomass burning emissions (labeled as FFBB and FF, respectively), have
been conducted to investigate the impacts of biomass burning aerosols on convective
systems in the Maritime Continent through both direct and indirect effects. Our study
focuses on the fire season from June to September of 2008. Therefore, the simulations
start from 1 May of 2008 and last for five months. The first month is used as a spin-up
period. Among the years with available emission data, both emission amount of biomass
burning and total precipitation in 2008 approximate their ensemble mean or represent an





average condition. Nevertheless, interannual variation of biomass burning emissions
alongside precipitation in the studies regions do exist (Lee et al., 2017; Lee et al., 2018),
and the influence of such variation on the effects of fire aerosol on convection should be
addressed in future studies.

## 2.3  Analysis methods

The primary target of this study is the convective systems associated with Sumatra
squall lines and diurnal rainfall over Borneo. Thus, our analyses mainly focus on the
convections over two specific regions: the Sumatra region (r1 in Fig. 1) and the Borneo
region (r2 in Fig. 1). The area coverage of the Sumatra region (r1) is from 97° to 103° E
in longitude and 0° to 6° N in latitude, while the area coverage of the Borneo region (r2)
is from 109° to 115° E in longitude and 1° S to 5° N in latitude.
To examine the impacts of fire aerosols on cloud formation and rainfall intensity as
well as amount, we have selected three convective systems each for the two focused
regions to perform an in-depth case study. We first trace the path of individual
convections and focus the analyses on the specific area of each of these convective
systems to identify the impacts of fire aerosols. Table 1 shows the selected cases in the
Sumatra region (r1) and the Borneo region (r2).
The consequent analyses are then focused on the fire-season-wise statistics of
convections for each study region. Table 2 shows the fire periods in the two study
regions. There are total of 54 convective systems simulated during the fire periods in the
Sumatra region (r1) and 35 convective systems in the Borneo region (r2).



The statistical quantities used in this study follows Wang (2005) to estimate the
mean value over a specific region (e.g., r1 or r2). The cloud area mean quantities are
defined as a function of output time step (t) by the following equation:
$$\bar{c}^{area}(t) = \frac{1}{N(t)}\sum_{\substack{q>qmin \\ n>nmin}} c(x,y,z,t).$$
(1)

Here $c$ is a given quantity (e.g., cloud water mass). Eq. (1) only applies to the grid points
where both the mass concentration $q$ and number concentration $n$ of a hydrometeor
exceed their given minima. The total number of these grid points at a given output time
step $t$ is represented by $N(t)$. The cloud area mean quantities are used to present the
average quantities of a given variable at a given output time step. Note that the cloud
area mean quantities only apply to hydrometeors. For rainfall, the analyzed quantities are
spatial averages over a specific area of the convective system for case study or over the
entire study region for longer-term statistic estimate.
# 3   Results
## 3.1  Model evaluation
### 3.1.1 Precipitation
The satellite-retrieved precipitation of the Tropical Rainfall Measuring Mission
(TRMM) 3B42 3hrly (V7) dataset (Huffman et al., 2007) is used in this study to evaluate
simulated rainfall. Figure 2a and 2b show the Hovmöller plots of daily TRMM and
FFBB precipitation from 1 June 2008 to 30 September 2008, respectively. Compared to
the satellite-retrieved data, the model has captured all the major rainfall events in the two
analysis regions (Fig. 3). In addition, because of its higher spatial resolution than



TRMM, the model produces more light rain events. Nevertheless, as indicated in our
previous study (Lee et al., 2017), a wet bias of the model is evident and mainly comes
from water vapor nudging in data assimilation (FDDA). As a result, the daily average
rainfall in FFBB over the Sumatra region (r1) is $11.05\pm5.90$ mm day$^{-1}$ from 1 June 2008
to 30 September 2008, higher than that of $7.21\pm5.54$ mm day$^{-1}$ derived from TRMM
retrieval. The wet bias also exists in the modeling results in the Borneo region (r2),
where daily average rainfall there is $15.40\pm8.49$ mm day$^{-1}$ in FFBB and only $9.56\pm7.20$
mm day$^{-1}$ in TRMM. For the simulated rainfall in FFBB, the temporal correlation with
TRMM is 0.44 in the Sumatra region (r1) and 0.64 in the Borneo region (r2).

## 244     3.1.2 Aerosol optical depth (AOD)

Because of limited ground-based observational data of aerosols, we use Aerosol
Optical Depth (AOD) from the level-3 Moderate Resolution Imaging Spectroradiometer
(MODIS) gridded atmosphere monthly global joint product (MOD08_M3;
http://dx.doi.org/10.5067/MODIS/MOD08_M3.061) to evaluate modeled aerosol spatial
distribution and relative concentration. Figure 4a shows MODIS monthly AOD in
Southeast Asia in September 2008. High AOD occurs in the southern part of Sumatra
and the southwestern part of Borneo. Compared to the MODIS retrieval, the modeled
AOD in FFBB has similar spatial distribution but a higher value (Fig. 4b). It is because a
high spatiotemporal resolution in our simulation enables the model to capture episodic
fire events better. In contrast, FF simulation produces much lower AOD values than
those of MODIS and FFBB, thus suggesting biomass burning aerosols make a substantial
fraction in atmospheric AOD during burning seasons.





### 3.1.3 Sounding profiles


We have used multiple weather sounding profiles measured in Bintulu Airport,
Malaysia (113.03° E, 3.20° N), provided by University of Wyoming
(http://weather.uwyo.edu/upperair/sounding.html).  An example for detailed summary is
a case at 12 UTC on 22 September 2008 (Fig. 5a).  This sounding provides information
of atmospheric state (e.g., vertical distributions of pressure, temperature, wind speed,
wind direction, and humidity) coinciding with one of our selected case study (r2c3) of
diurnal convective rainfall in Borneo.  Compared to the observed sounding data, the
FFBB simulation has produced similar temperature and wind profiles and well captured
the low-level and high-level wind speeds and wind directions (Fig. 5a versus 5b).  It also
well predicts several key indexes of convection: temperature and pressure of the Lifted
Condensation Level (LCL) simulated in FFBB are 296.2 K and 955 hPa, respectively,
which are close to the values of 296.2 K in temperature and 960.7 hPa in pressure derived
from the observed sounding data.  The model predicts 3049 J of Convective Available
Potential Energy (CAPE), while 2031 J of CAPE is estimated in the observed sounding
data.  Besides this 22 September 2008 case, the model has also captured major features of
observed profiles for all the other cases selected in our analyses.

### 3.1.4 Cloud vertical structure


The Cloud-Aerosol Lidar and Infrared Pathfinder Satellite Observation (CALIPSO)
provides information of the vertical structure of clouds on its path around the globe
(https://www-calipso.larc.nasa.gov/products/lidar/browse_images/production/),  including
that of one of our cases (r2c3) of diurnal convective rainfall in Borneo on September 22,



2008 (Fig. 6a).  For this case, CALIPSO shows the vertical structure of a convective
system over Borneo along with high $PM_{2.5}$ concentration near the surface (yellowish
color near the surface), implying a potential impact of biomass burning aerosols on
convective clouds.  It can be seen that the FFBB simulations well captures the vertical
structure of convective clouds as well as the near-surface aerosol layers, including their
vertical extension (Fig. 6c versus 6a).  With the comparison of FF simulation, we are able
to identify the biomass burning origin of these aerosols near the surface.

## 3.2  Analyses of selected cases in two study regions

### 3.2.1 The Sumatra region (r1)

The three selected cases in r1 or the Sumatra region (r1c1, r1c2 and r1c3) all
occurred in the afternoon (2 PM or 5 PM local time) and lasted less than 24 hours (Table
1).  Most fire aerosols in this study region were initially emitted from the central and
south Sumatra then transported along with southwesterly winds to encounter convections
in the northern Sumatra.  Compared to the result of FF, $PM_{2.5}$ concentration in FFBB can
be 6~12 times higher in the Sumatra region (r1) in these selected cases (Fig. 7).
Aerosols from biomass burning in FFBB add 2~3 times more cloud droplet number
concentration and 8~20% higher cloud water mass compared to the results in FF (Table
2).  The mean radius of cloud droplets in FFBB is about 6~7 μm, clearly smaller than that
in FF (10~11 μm).   Smaller cloud droplet in FFBB reduces the efficiency of
autoconversion, and further decreases rain water mass and raindrop number
concentration.  Hence, raindrop number concentration in FFBB is 40~50% lower than
that in FF among our selected cases in r1 (Table 3).  However, besides autoconversion,


rain water mass is also affected by other microphysics processes.  Larger raindrops
combining smaller cloud droplets in FFBB can increase the efficiency of cloud droplet
collection by rain and thus produce higher rain water mass but number, possibly
compensating the decrease of rain water mass resulted from lowered autoconversion.
Overall, rain water mass decreases 15% in the case of r1c2 and 10% in the case of r1c3,
respectively.  Compared to the cases of r1c2 and r1c3, the case of r1c1 is a relatively
weak convective system.  After introducing fire aerosols, the mass concentration of snow
and graupel in this case increases 62% and 48%, respectively.  Melting snow and graupel
in the lower atmosphere results in a significant increase of rain water mass concentration
by 49%.  Thus, total hydrometeor mass is increased by 36% in FFBB from that in FF.
Note that the "aerosol-aware" microphysics scheme in WRF-Chem only applies to the
warm cloud process (Morrison et al., 2005; Morrison et al., 2009); therefore, ice
nucleation is parameterized of ambiance temperature only regardless of the aerosol
concentration.

In the FF simulations, the convective system in the case of r1c2 and r1c3 is stronger

than the system in the case of r1c1, and the average rainfall of r1c2 and r1c3 is also
higher than the rainfall of r1c1 (Table 4).  Adding fire aerosols in FFBB does not
substantially change the average rainfall in r1c2 and r1c3 (+3% and -8%, respectively;
Table 4).  However, in the relatively weak convective system of r1c1, adding fire
aerosols significantly increases the mean rainfall amount by 106% ($1.33\pm0.47$ mm $3hr^{-1}$
in FF versus $2.74\pm1.21$ mm $3hr^{-1}$ in FFBB).





## 3.2.2 The Borneo region (r2)

The three selected cases in r2 (r2c1, r2c2, and r2c3) also occurred during the summer monsoon season when active biomass burning events existed in the west Borneo. In these cases, fire aerosols were transported to the north and northeast by the southeasterly and southwesterly winds. Because of the proximity of fire emissions, the $PM_{2.5}$ concentration in FFBB can be 24 times higher than that in FF in the Borneo region (r2) in these selected cases (Fig. 7).

The modeled results demonstrate the substantial impacts of fire aerosols on both ambient aerosol concentration and cloud droplet number concentration. $PM_{2.5}$ concentration in FFBB is drastically higher than that in FF with the highest increase appears in the case of r2c1 at 4940%, more than doubled the values of r2c2 (2402%) and r2c3 (2422%). The increase in cloud droplet number concentration in the case of r2c1 (703%) is also substantially higher than those in r2c2 (337%) and r2c3 (409%) (Table 2). The mean radius of cloud droplets in FFBB is about 6~7 μm, which is substantially smaller than that in FF (10~11 μm). The mean cloud droplet radii in FF and FFBB in r2 are similar to the results in r1. On the other hand, the increase of cloud water mass due to fire aerosols is not so dramatic in all these cases, only about 8%~27% higher than that in the FF simulations (Table 3). As discussed above, rain number concentration in FFBB over the Borneo region (r2) is lower than that in FF, similar to the cases in r1, likely due to the low efficiency of autoconversion induced by the presence of a large quantity of smaller cloud droplets. Rain water mass of FFBB in the r2c1 case is decreased by about 6% due to fire aerosols, which is similar to the results in the r1c2 and r1c3 cases over the Sumatra region (Table 3). However, interestingly, rain water and snow mass are both





substantially increased in FFBB by 64% and 69% in r2c2 and by 19% and 60% in r2c3,
respectively (Table 3). The cases of r2c2 and r2c3 are relatively weak convective
systems, similar to the case of r1c1. Our results show that fire aerosols have substantial
impacts on cold cloud processes in the weak convective systems. Overall, total
hydrometeor mass concentration in FFBB have increased 47% in r2c2 and 13% in r2c3.
The changes of rainfall amount due to fire aerosols in r2 are similar to the cases in r1.
For the strong convection case of r2c1, adding fire aerosols in the FFBB simulation
decreases the total rainfall amount by 18%. However, in the weak convection cases of
r2c2 and r2c3, adding fire aerosols would double the rainfall amount (Table 4).
Compared to the results in FF, rainfall intensity is persistently higher in FFBB during the
convection life cycle in those weak convection cases. Nighttime rainfall intensity in
FFBB, especially, is much higher than the rainfall intensity in FF. Therefore, as shown
by our results, fire aerosols appear to have more substantial impacts on the quantities of
hydrometeors and rainfall of the weak convection cases in both Sumatra region (r1) and
Borneo region (r2).
## 3.3 Fire-season statistics of convections in two study regions
Statistics covering the entire simulated fire season (~4 months) for each study region
have been derived to provide trend/tendency information regarding several aspects of the
impact of fire aerosols on convections. In our simulations, $PM_{2.5}$ concentration in FF
during the fire periods, which can be regarded as the background value for FFBB
simulation before adding fire aerosols, is $1.36\pm0.19$ μg m$^{-3}$ in r1 and $0.56\pm0.09$ μg m$^{-3}$ in
r2. In comparison, $PM_{2.5}$ concentration in FFBB is $11.37\pm10.41$ μg m$^{-3}$ in r1 and
$10.07\pm7.73$ μg m$^{-3}$ in r2.  Note that unlike in some other studies where the control
simulations use constant aerosol concentrations, fire aerosol concentrations in our
simulations can vary in responses to changes in fire emissions, or aerosol removal by rain
scavenging due to precipitation change caused by fire aerosols themselves.  Hence, the
processes included in our simulations are closer to reality, and the results could better
reflect the nature of fire aerosol-convection interaction in the Maritime Continent.

Averaged through the entire modeled fire periods, cloud water mass (Qc), cloud

droplet number concentration (Qnc), and rain drop number concentration (Qnr) in FFBB
differ substantially from those in FF, demonstrating the influence of fire aerosols.  Figure
8 shows that adding fire aerosols in FFBB would increase Qc by 14% and Qnc by 226%
in r1, and Qc by 18% and Qnc by 349% in r2.  Another pronounced change in response to
adding fire aerosols is a decrease in Qnr by 44% in r1 and 47% in r2.  Although an
increase in snow mass (Qs) and graupel mass (Qg) and a decrease in rain water mass (Qr)
after adding fire aerosols, the uncertainty of these hydrometeor changes is large.

In Sect. 3.2, we have discussed the significant rainfall increase occurred in the weak

convective systems after adding fire aerosols.  Here we use the fire-season statistics to
further this discussion.  Regardless the strength of convective precipitation, the mean 3-
hourly rainfall during the fire periods is $1.06\pm0.85$ mm 3hrs$^{-1}$ in FF and $1.09\pm0.86$ mm
3hrs$^{-1}$ in FFBB over the Sumatra region (r1), statistically does not change.  The rainfall
difference in the Borneo region (r2) between FF and FFBB is also insignificant
($1.32\pm1.20$ mm 3hrs$^{-1}$ in FF versus $1.35\pm1.14$ mm 3hrs$^{-1}$ in FFBB).

On the other hand, based on the diurnal rainfall pattern in two study regions, we

notice that daily maximum and minimum rainfall show apparent differences between the

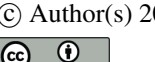



FFBB and FF simulations in r2, while such differences are rather small in r1 (Fig. 9).
The maximum or minimum rainfall intensity in the two simulations are closely aligned
with the 1:1 line in Fig. 9a and 9b.  However, when looking into each of the 54
convective events in r1, there are 30 events where the model predicted higher maximum
and minimum rainfall intensity in FFBB than in FF.  These are mostly weak convective
events.

Additionally, and somewhat opposite to the rainfall statistics in r1, the intensity of

maximum and minimum rainfall in r2 is higher in FF than in FFBB.  The daily rainfall
peak in r1 is mostly less than 3 mm 3hrs$^{-1}$; in comparison, one-third of convective events
in r2 have daily maximum rainfall exceeding 3 mm 3hrs$^{-1}$.  We have categorized the
maximum rainfall based on its values in the afternoon and midnight.  We find that those
heavy maximum rainfalls in r2 tend to occur in the midnight (Fig. 9c) associated with the
anticyclonic circulation formed in the western Borneo induced by southeasterly winds
from the Southern latitude turn northeastward along the west coast of Borneo, owing to
the terrain of Borneo Island and the sea breezes from the South China Sea.  The vortex
produced by such a circulation leads to strong updraft and then strong convection.   Note
that this anticyclonic circulation is different from the Borneo vortex, the latter appears as
a persistent feature of the boreal winter climatology and is related to the northeasterly
from the South China Sea and cold surge events (Chang et al., 1983; Chang et al., 2005).

The low-level wind pattern of Borneo convections is similar to the westerly regime,

especially the weak westerly (WW) regime identified by Ichikawa and Yasunari (2006).
According to their analysis, the WW regime tends to occur in boreal summer. Its
composites include an anticyclonic feature with the weak wind field over the Borneo





Island. The deep convective storms developed in the WW regime tend to stay close to
the west coast associated with the lower-level convergence enhanced by the prevailing
wind and local circulations around there, resulting in localized rainfall over the offshore
region of the west coast. Based on our simulations, the onset of convection occurs in the
afternoon over the western mountain range of Borneo. These storms would consequently
evolve into widespread shallow storms in the evening over the western part of the island.
The maximum rainfall appears on the west coast because of a local westward propagating
rainfall system that develops around midnight or early morning.

The comparison of the maximum rainfall between FF and FFBB in Fig. 9 shows that

fire aerosols tend to reduce the maximum rainfall, especially for high-intensity rainfall
events. In other words, fire aerosols have substantial impacts on the nocturnal
convections, which are associated with the local anticyclonic circulation in the western
Borneo. This effect on nocturnal convections in the western Borneo by fire aerosols will
be discussed further in the next section.

### 3.4  The impact of biomass burning activities on nocturnal

### 428    convections in the Borneo region

To further analyze the effects of fire aerosols on nocturnal convections, we have

categorized convective events into nocturnal convections (NC) and non-nocturnal
convections (non-NC), based on whether the maximum rainfall occurs from midnight to
early morning or in the time frame from late afternoon to evening. Figure 10 shows the
diurnal time series of precipitation averaged over the Borneo region (r2) in FF and FFBB.





Again, 3-hour-mean rainfalls of nocturnal convections are higher than those of non-
nocturnal convections in both simulations.

Nocturnal convections tend to stay close to the west coast associated with a lower-

level convergence enhanced by the prevailing wind and local circulations mainly related
to the land breezes from inland of the western Borneo. The strong convergence near the
surface over the offshore region of the west coast causes the weak westerly monsoon
windflaws and local land breezes to merge during the nighttime. However, during the
fire periods, the daytime absorption of fire aerosols (e.g., black carbon) can cause an
atmospheric warming (even without fire generated heating flux being incorporated in the
model). This could increase near surface air temperature, weaken land breezes and thus
surface convergence. As a result, the nocturnal convections in FFBB cannot develop as
strong as those in FF. On the other hand, both nocturnal and non-nocturnal convections
are initiated over the western mountain range under a prevailing wind of the sea breezes
from the South China Sea. The increases of near surface temperature owing to the fire
aerosols can enhance this prevailing wind from the ocean and thus lead to a higher
convective rainfall in FFBB during the onset stage of the nocturnal convections as well as
non-nocturnal convections.

Diurnal evolution of vertical profiles clearly indicates that mass mixing ratio of total

hydrometeors, temperature, and vertical velocity differ in both daytime and nighttime
between FF and FFBB for those nocturnal convections (Fig. 11). The differences of near
surface temperature between FF and FFBB are more pronounced during the period after
sunset (Fig. 11d). The differences of near surface temperature mainly happen over land,

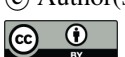



and the higher near surface temperature in FFBB weakens the land breezes and near
surface convergence along the coast.   Starting from late afternoon, (about 5 PM local
time), vertical velocity increases with time until sunrise next day in both simulations (Fig.
11e) due to the convergence of the monsoon windflaws and local land breezes during the
nighttime, and this matches very well with that of mass mixing ratio of total
hydrometeors (Fig. 11a and 11e).   Noticeably, the main differences in vertical velocity
and hydrometeor mass mixing ratio between FFBB and FF also start to become evident
after entering the evening.   Because of the weaker convergence near the surface in FFBB,
the differences in vertical velocity at the higher altitude between FFBB and FF peaks in
the nighttime.

It should be indicated that if the heat flux generated by fires was incorporated in the

model, the warming effects from biomass burning would be even stronger and could
persist in nocturnal timeframe as demonstrated in Zhang et al. (2019).   However, this
would likely be more effective for open fire regime.   For most of peat fires, burning is
largely proceeded underground.   Based on our significantly reduced heat flux for the peat
fires as discussed in Sect. 2.1, if the heat flux was incorporated in the model, such fires
would not increase surface temperature by 4-5 °C as suggested for the tropical (open) fire
cases in Zhang et al. (2019).

As a summary, the schematics shown in Fig. 12 illustrate the impact of biomass

burning activities on nocturnal convections in the Borneo region.   In the daytime, under
the prevailing wind of sea breezes from the South China Sea, convections develop over
the western mountain range.   Because near surface heating from the absorption of



sunlight by fire aerosols could enhance the prevailing wind from the ocean, convective
rainfall becomes higher at the onset stage of the nocturnal convections (still in daytime)
due to biomass burning activities (Fig. 12b).  In the nighttime, convection moves to the
offshore region of the western Borneo.  The strong convergences near the surface merge
the weak westerly monsoon windflaws with local nighttime land breezes to form an
anticyclonic circulation (Fig. 12c).  During the fire periods, the daytime near surface
warming by fire aerosols could also further weaken land breezes and surface
convergence.  Hence, the nocturnal convections during fire events would not develop as
strong as in days without fires (Fig. 12d versus 12c).
**4   Summary**

By comparing WRF-Chem modeling results include or exclude biomass burning

emissions (FFBB versus FF), we have identified certain detailed impacts of fire aerosols
on convective events within two study regions in the Maritime Continent during a four-
month period (June 2008 ~ September 2008).  In total, 54 convective systems in the
Sumatra region and 35 convective systems in the Borneo region have been simulated.
Three convective events of each study region have been selected for in-depth
investigation.  In addition, statistical analyses have been performed throughout the entire
simulation period for each region.  We have focused our analyses on two rainfall
features: 1) convective precipitation associated with Sumatra squall lines, and 2) diurnal
rainfall over the western Borneo.

We find that fire aerosols lead to the increase of cloud water mass and cloud droplet

number concentration among all analyzed cases while a substantial reduction of rain drop
number concentration.  Influences of fire aerosols on other hydrometeors vary from case



to case. Specifically, our results show that fire aerosols can significantly change the
quantities of hydrometeors, particularly those involved in cold cloud processes and
rainfall of weak convections in either the Sumatra region or the Borneo region. Rainfall
intensity is higher in FFBB during the entire convection life cycle in those weak
convection cases, and the nighttime rainfall intensity in FFBB is significantly higher than
that in FF.
Statistics performed throughout the entire modeled fire season shows that the fire
aerosols only cause a nearly negligible change (2-3%) to the total rainfall of convective
systems in both study regions. On the other hand, we notice that fire aerosols can still
alter daily maximum and minimum rainfall in some cases, for example, fire aerosols lead
to the increase of maximum and minimum rainfall intensity in 30 weak convective events
in the Sumatra region.
In the Borneo region, biomass burning activities mainly affect the rainfall intensity
of nocturnal convection. Because near surface heating from the absorption of fire
aerosols can enhance the prevailing wind from the ocean (sea breeze) during the daytime,
the convective rainfall over the western mountain range is higher during the onset stage
of the nocturnal convections. In the nighttime, the consequence of the above
thermodynamic perturbation by absorbing fire aerosols can further weaken land breeze
and surface convergence. Hence, the rainfall intensity of nocturnal convections under the
influence of fire aerosols would become weaker.
This study has demonstrated how biomass burning activities could affect convective
systems in the Maritime Continent by altering cloud microphysics and dynamics. We
find the biomass burning activities significantly change the diurnal rainfall intensity,





especially those low-level wind patterns associated with the weak westerly (WW) regime
as suggested by Ichikawa and Yasunari (2006). Our results show that neither a single
case study nor a simple statistical summary applied to overall model simulation period
without in-depth analyses could reveal the impact of biomass burning aerosols on
convections under different windflaw regimes.

## 529   Data availability

FINNv1.5 emission data are publicly available from
http://bai.acom.uar.edu/Data/fire/. REAS emission data can be downloaded from
https://www.nies.go.jp/REAS/. TRMM data can be obtained from
https://pmm.nasa.gov/data-access/downloads/trmm. AOD from MODIS can be
obtained from http://dx.doi.org/10.5067/MODIS/MOD08_M3.061. Sounding profiles
are publicly available on http://weather.uwyo.edu/upperair/sounding.html. WRF-Chem
simulated data are available upon request from Hsiang-He Lee (lee1061@llnl.gov).

## 537   Author contribution

H.-H. L. and C. W. designed the experiments and H.-H. L. carried them out. H.-H.
L. configured the simulations and analyzed the results. H.-H. L. and C. W. wrote the
manuscript.

## 541   Acknowledgments

This research was supported by the National Research Foundation Singapore through
the Singapore-MIT Alliance for Research and Technology, the interdisciplinary research



program of Center for Environmental Sensing and Modeling. It was also supported by
the U.S. National Science Foundation (AGS-1339264) and L'Agence National de la
Recherche (ANR) of France through the Make-Our-Planet-Great-Again Initiative, ANR-
18-MPGA-003 EUROACE. The authors would like to acknowledge NCEP-FNL and
NCAR FINN working groups for releasing their data to the research communities; and
the NCAR WRF developing team for providing the numerical model for this study. The
computational work for this article was performed on resources of the National
Supercomputing Centre, Singapore (https://www.nscc.sg).







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


Table 1. The case period of the selected cases in the Sumatra region (r1) and the Borneo
region (r2)

| Case name | Case period |
|-----------|-------------|
| r1c1 | 2008/08/10 0900 UTC ~ 2008/08/11 0300 UTC |
| r1c2 | 2008/08/19 0600 UTC ~ 2008/08/20 0000 UTC |
| r1c3 | 2008/09/23 0900 UTC ~ 2008/09/24 0000 UTC |
| r2c1 | 2008/08/05 0900 UTC ~ 2008/08/06 0300 UTC |
| r2c2 | 2008/09/17 0600 UTC ~ 2008/09/17 2100 UTC |
| r2c3 | 2008/09/22 0300 UTC ~ 2008/09/23 0000 UTC |




Table 2. The fire periods in the two study regions

| The Sumatra region (r1) | The Borneo region (r2) |
|---|---|
| 6/10/2008 ~ 6/20/2008 | 6/21/2008 ~ 6/27/2008 |
| 6/25/2008 ~ 6/28/2008 | 8/1/2008 ~ 8/8/2008 |
| 7/4/2008 ~ 7/7/2008 | 9/10/2008 ~ 9/30/2008 |
| 7/27/2008 ~ 8/20/2008 | |
| 9/17/2008 ~ 9/27/2008 | |






Table 3. The mean differences in percentage of FFBB to FF (i.e. (FFBB-FF)/FF × 100%)
for each selected case over the main convection area in the Sumatra region (r1) and the
Borneo region (r2). Qc, Qi, Qr, Qs and Qg represents cloud, ice, rain, snow, and graupel
mass concentration respectively.  Qnc, Qni, Qnr, Qns and Qng means number
concentration for each hydrometeor.

| Case | Qc | Qi | Qr | Qs | Qg | Qnc | Qni | Qnr | Qns | Qng |
|------|-----|-----|------|------|-----|------|-----|------|------|------|
| r1c1 | 8% | 27% | 49% | 62% | 48% | 248% | 55% | -41% | 33% | 39% |
| r1c2 | 20% | -6% | -15% | -25% | 1% | 349% | -1% | -45% | -11% | -6% |
| r1c3 | 18% | 10% | -10% | 3% | 5% | 311% | 4% | -50% | 11% | -6% |
| r2c1 | 27% | 1% | -6% | -5% | -4% | 703% | 3% | -59% | 4% | -5% |
| r2c2 | 22% | 10% | 64% | 69% | 58% | 337% | 24% | -32% | 17% | 57% |
| r3c3 | 8% | 10% | 19% | 60% | -2% | 409% | -5% | -66% | 8% | -12% |






Table 4. The averaged precipitation (mm 3hrs$^{-1}$) of FFBB and FF for each selected case
over the main convection area in the Sumatra region (r1) and the Borneo region (r2).
Parentheses in the third column show the difference in percentage of FFBB to FF (i.e.
(FFBB-FF)/FF × 100%).

| Case | FF | FFBB |
|------|------|------|
| r1c1 | 1.33±0.47 | 2.74±1.21 (+106%) |
| r1c2 | 2.97±1.42 | 3.05±1.49 (+3%) |
| r1c3 | 4.32±1.84 | 3.98±2.18 (-8%) |
| r2c1 | 3.73±2.64 | 3.07±1.21 (-18%) |
| r2c2 | 1.88±0.53 | 3.97±1.47 (+111%) |
| r3c3 | 0.54±0.53 | 1.10±1.02 (+103%) |




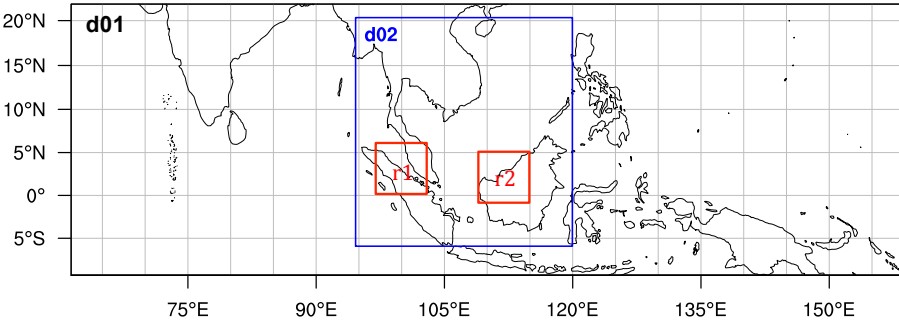

Figure 1. Domain configuration for WRF-Chem simulations. Domain 1 (d01) has a
resolution of 25 km, while Domain 2 (d02) has a resolution of 5 km. Two red boxes
indicate the two study regions: the Sumatra region (r1) and the Borneo region (r2).



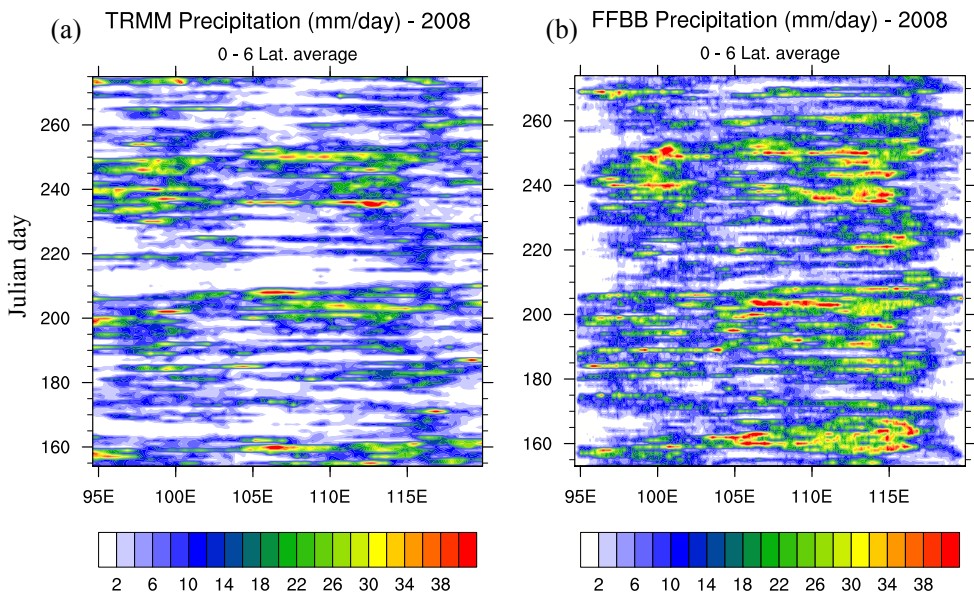

Figure 2. Hovmöller (time versus longitude) plot of daily precipitation (mm day$^{-1}$) from 1
June 2008 to 30 September 2008 from:(a) Tropical Rainfall Measuring Mission (TRMM)
and (b) FFBB. Latitude average is from 0° to 6°N.



**(a) Rainfall comparison – r1**

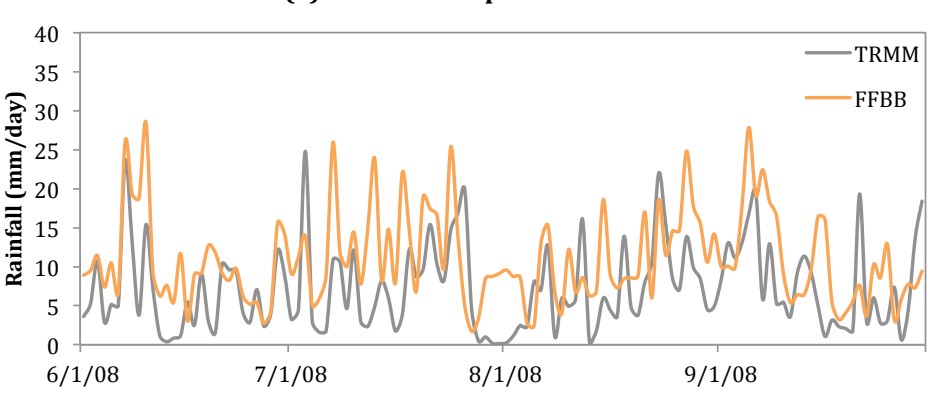


**(b) Rainfall comparison – r2**

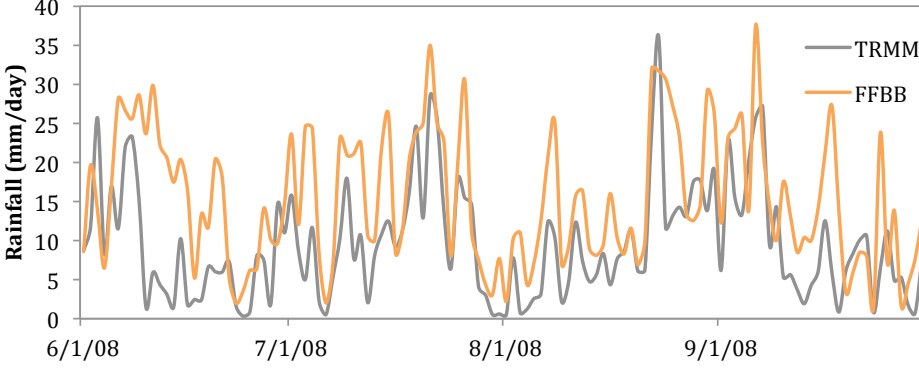

Figure 3. Time series of area-averaged daily rainfall (mm day$^{-1}$) from Tropical Rainfall
Measuring Mission (TRMM) and FFBB over (a) the Sumatra region (r1) and (b) the
Borneo region (r2).





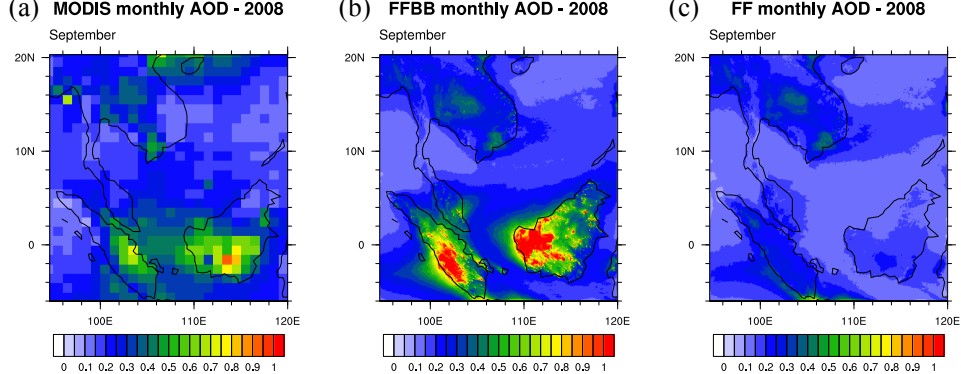

Figure 4. Monthly aerosol optical depth (AOD) in September 2008 from (a) Moderate Resolution Imaging Spectroradiometer (MODIS), (b) FFBB, and (c) FF.





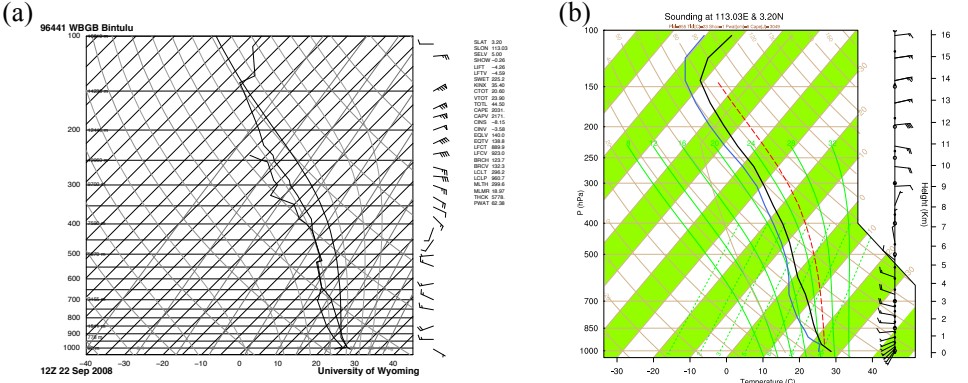

Figure 5. (a) Sounding profile observed at Bintulu Airport, Malaysia (113.03° E, 3.20° N)
at 12 UTC on 22 September 2008. (b) Modeled sounding profile in FFBB at the same
location and time as (a).





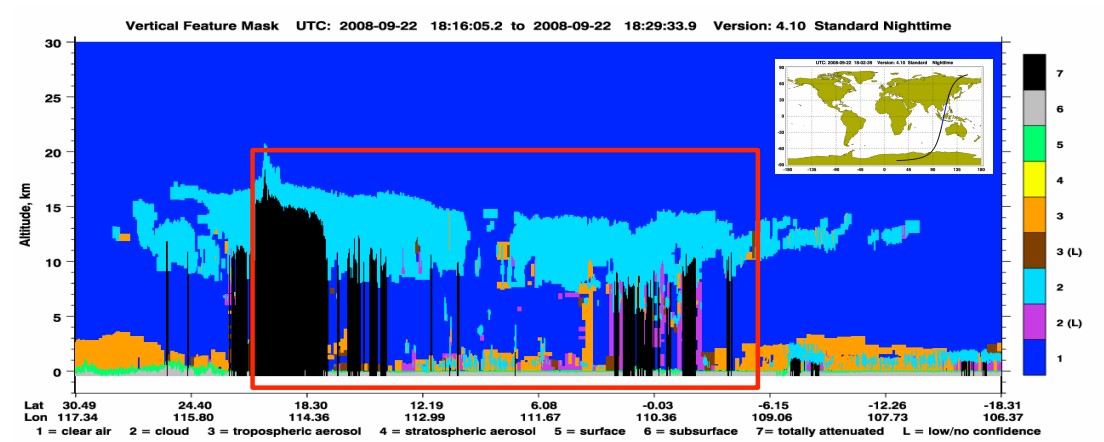


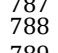

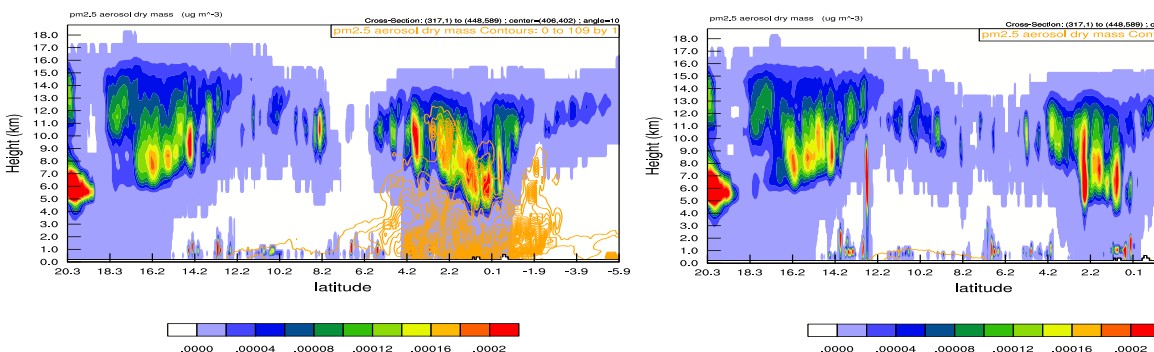






Figure 6 (a) The vertical structure of cloud retrieved from the Cloud-Aerosol Lidar and Infrared Pathfinder Satellite Observation (CALIPSO) on September 22, 2008. (b)-(c) The sum of simulated hydrometeor mixing ratio (shaded; kg kg$^{-1}$) and PM$_{2.5}$ concentration (contour; μg m$^{-3}$) in FFBB and FF, respectively. The profile domain of (b) and (c) is corresponding to the red rectangle in (a).





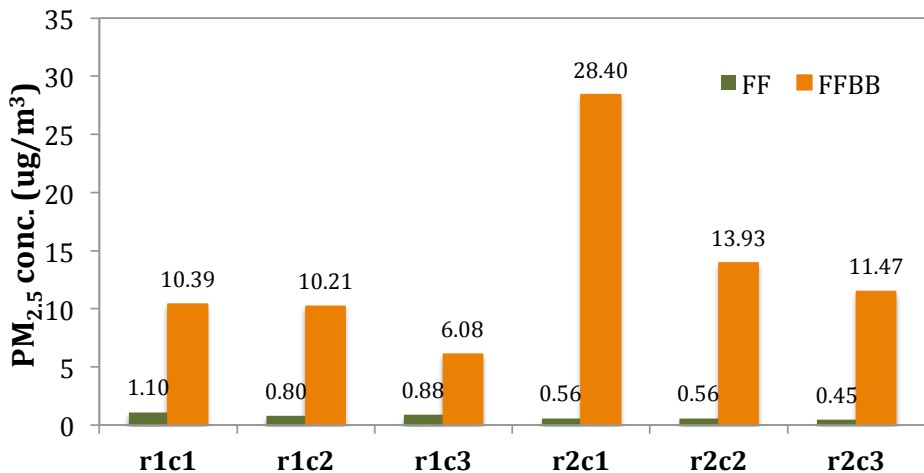

Figure 7. The mean PM$_{2.5}$ concentration (μg m$^{-3}$) in FF and FFBB for selected cases in the
Sumatra region (r1) and the Borneo region (r2).




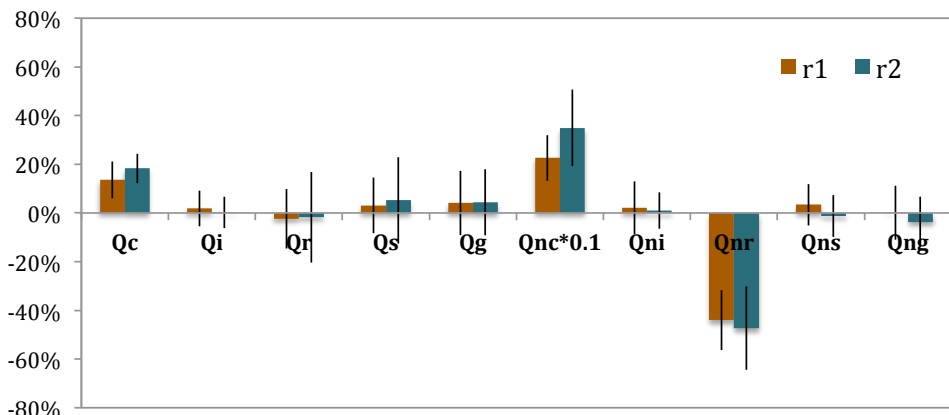

Figure 8. The mean differences in percentage of FFBB to FF (i.e. (FFBB-FF)/FF × 100%)
over all convective cases during the fire periods in the Sumatra region (r1) and the Borneo
region (r2). Qc, Qi, Qr, Qs and Qg represents cloud, ice, rain, snow, and graupel mass
concentration, respectively. Qnc, Qni, Qnr, Qns and Qng means number concentration for
each hydrometeor. The error bars represent one standard deviation.



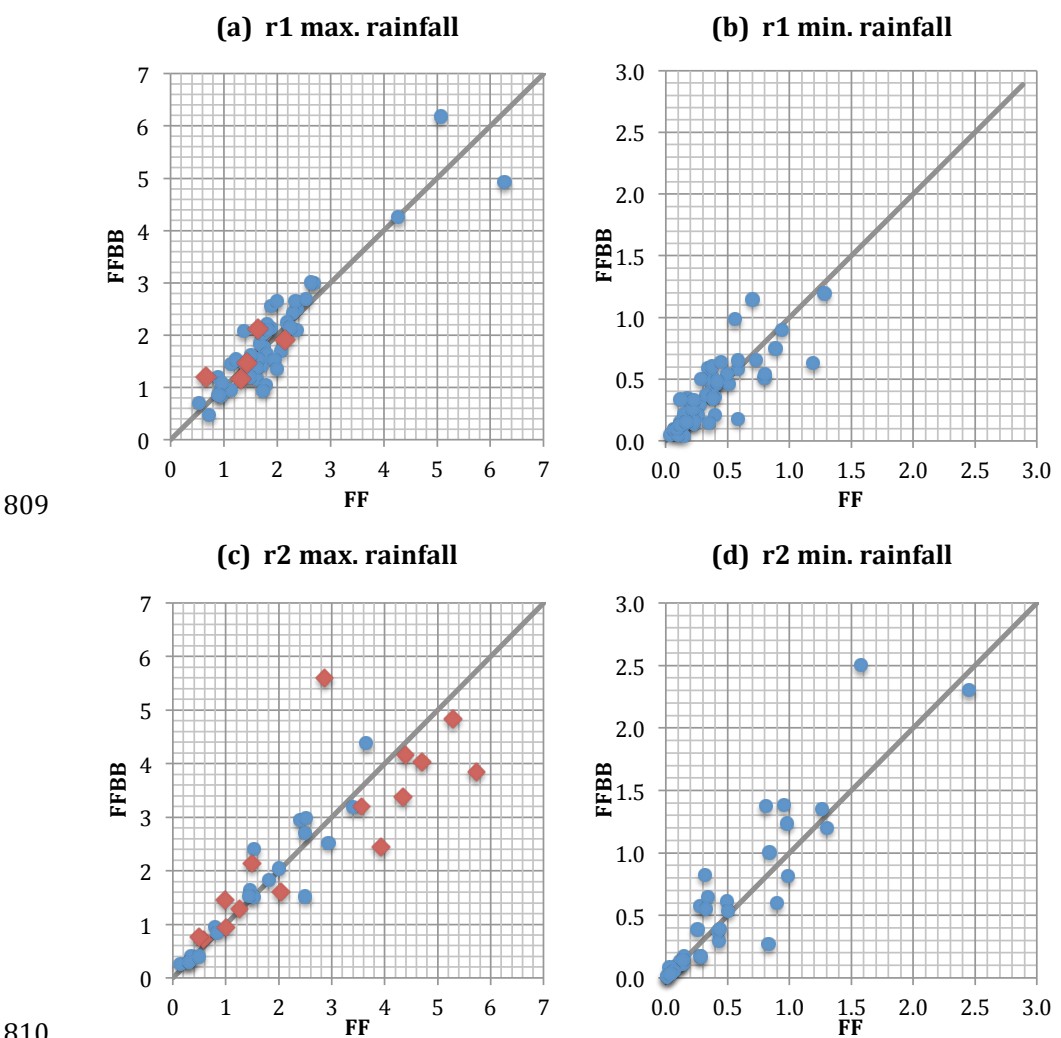


Figure 9. The scatterplots of daily maximum and minimum convective rainfall (mm 3hr$^{-1}$)
during the fire periods in in the Sumatra region (r1) and the Borneo region (r2). Red
diamonds in (a) and (c) indicate that the maximum convective rainfall conducts in the
midnight or early morning.


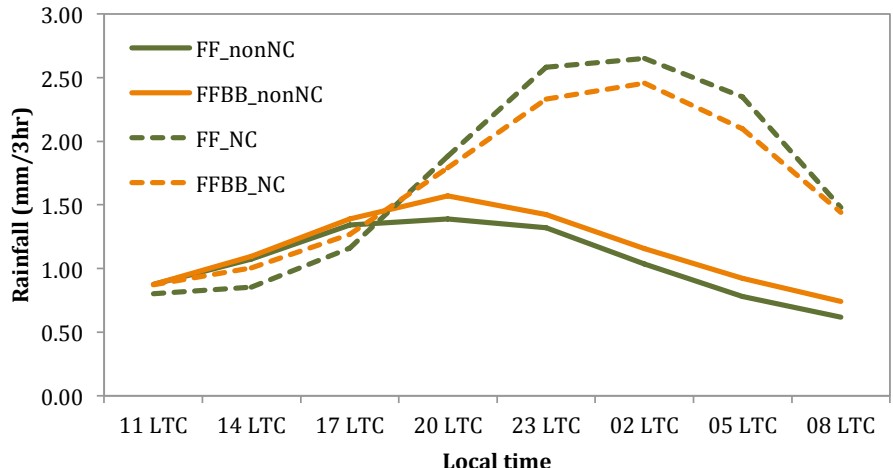

Figure 10. The diurnal time series of precipitation averaged over the Borneo region (r2) for
nocturnal convections (NC) and non- nocturnal convections (non-NC) during fire periods in
FF and FFBB.




Figure 11. Diurnal evolution of vertical profiles over the Borneo region (r2) in FF for (a)
total hydrometeor mixing ratio (mg kg$^{-1}$), (c) temperature (°C), and (e) vertical velocity (m s$^{-1}$). Data are averaged all the nocturnal convections. (b), (d), and (f) is the differences
between FF and FFBB (FFBB-FF) for each parameter.





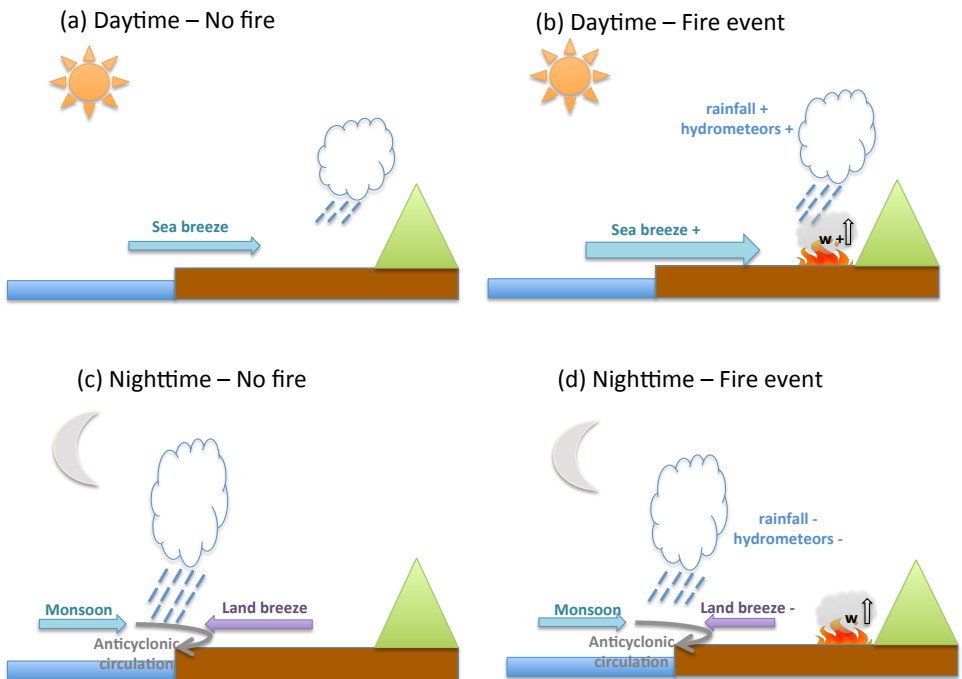

Figure 12. Schematics of diurnal rainfall/convection activity over the western Borneo. (a)
and (b) illustrate the formation of convection during the daytime without and with fire event,
respectively. (c) and (d) are the same as (a) and (b) but in the nighttime.