# Peer review of "The Impacts of Biomass Burning Activities on Convective Systems in the Maritime Continent"

_Atmospheric Chemistry and Physics, 2019_

## Referee Comment (RC1) · Anonymous Referee #2 · 18 Sep 2019

This paper examines the impacts of biomass burning aerosols on convective systems over the northern Sumatra and the western Borneo in the Maritime Continent based on based on long-term WRF-Chem simulations. While the paper is well written and interesting, there are some concerns that need to be addressed before the paper being publishable. (1) The resolution of inner domain at 5 km is still too coarse for simulating convective clouds. (2) The section of selected cases analysis looks vague and needs more detailed analysis. The impacts of aerosol on precipitation are very complicated (different mechanisms on different clouds under different conditions). It is too bold to get the conclusion just based on three cases even with cloud types unknown. There are so many convective cases that could be categorized them and analyzed in detail.

[Figure]

(3) The heating effect of fire aerosols seems too weak to have significant influence on circulation.

Here are also some specific comments:

1. Line 153: What is the time frequency of nudging?

2. Line 183: needless parenthesis

3. Line 205: How and why were these convective systems selected? Why only three?

4. Line 233: Were the model results interpolated to the resolution of TRMM before doing the comparison?

5. Line 260: Why only this example sounding is shown? You may compare with many other cases and even show a statistical comparison.

6. Line 275: Only one case captured by CALIPSO?

7. Line 311-314: It is confusing. Aerosol impact on ice-phase microphysical processes is still considered in Morrison through the CCN effect. It is the IN effect of aerosol that is missed.

8. Line 315-321: More background information of these cases are necessary. You just simply saying that one case has weaker convective systems than other two. This is too ambiguous.

9. Line 454: The temperature increase seems too small. Is this significant? Maybe the difference is within the model simulating error range.

10. Line 455-457: No figure showing this conclusion. How much land breeze and surface convergence is weakened.

---

## Referee Comment (RC2) · Anonymous Referee #3 · 19 Sep 2019

The authors attempt to investigate the fire aerosol-cloud-precipitation interactions by conducting modeling sensitivity studies. The performance of WRF-CHEM simulations were fully evaluated, and the responses of cloud microphysics and precipitation amount to fire aerosols were carefully quantified. However, I still have some minor issues about this work prior to its publication.

1. In the discussions in sections 3.1 and 3.2, it appears that the responses of cloud microphysics properties and precipitation to fire aerosols are sensitive to convection intensity of the systems selected for case studies, but the authors didn't show what are the criteria to determine the systems are convective weak. At least, some ba-

sic description about the selected convective systems should be provided so that the readership could have some sense about the convection strength of each system.

2. Related to point 1, is that possible to do statistical analysis of fire periods for weak and strong convective systems separately? Since the weak systems are more sensitive to fire aerosols, I would expect that there might be more significant differences in cloud properties or precipitation between fire aerosol case and non-fire aerosol case when looking at those weak systems.

3. Are the fire periods shown in Table 2 the time periods during which the fire aerosols are continuously emitted into atmosphere? I just want to make sure that the cloud systems selected for statistics of fire season are those which were indeed influenced by fire aerosols. That means the selected cloud systems for analysis concurrently occurred with fire events.

4. Please add uncertainties of precipitation for each case in Figure 9.

5. In section 3.4, the impacts of fire aerosols on local circulations like land/sea breeze are not evident. Some figures like the mean wind fields for fire aerosol and non-fire aerosol cases to show their difference would be helpful.

———————————————

---

## Referee Comment (RC3) · Anonymous Referee #1 · 20 Sep 2019

This study investigates the impact of biomass burning aerosols on convective systems in the Sumatra and Borneo regions of Southeast Asia using the WRF-Chem model. Considering the large uncertainty in the interactions between aerosols, particularly those from biomass burning, and convective clouds, this study advances our under-standing of the complicated and competing physical processes that governs the net effect of biomass-burning aerosols. The manuscript is generally well written. I think it can be considered for publication after the author addresses the following comments and suggestions. 1. Abstract: The descriptions after Line 45 are much too general. The author mentioned several times that fire aerosols have "significant/substantial impacts" on convection. What exactly are these impacts? I believe the author should summarize their main findings here so that the abstract can be more informative. 2. Line 173-175: How did you treat emissions from the flaming vs smoldering phases when calculating plume rise? A previous study (Shi et al., 2019, JGR-Atmospheres, DOI 10.1029/2019JD030472) has shown that the fraction of smoldering-phase emissions has a large impact on plume rise and fire-induced aerosol concentrations. 3. Line 185-186: I think it's not accurate to use "fossil fuel emissions" here. "Anthropogenic emissions" may be a better term. Many anthropogenic emissions do not originate from fossil fuels, such as VOC emissions from solvent use, $NH_3$ emissions from agricultural activities, and emissions from household biomass fuels. 4. Line 191-193: This is an important point. You may want to show the data in SI. 5. Section 3.1.2: Since fire emissions have a large day-to-day variability, I think the monthly mean AOD may not be suitable for evaluating the model performance. I suggest to use daily product (MOD08_D3) instead. Also, the author argues that the higher simulated AOD than observations is because "a high spatiotemporal resolution in our simulation enables the model to capture episodic fire events better". I think comparing with daily AOD observations could help to confirm whether this argument is true or not. 6. Line 303: but smaller number? 7. Line 307-308: Why do the mass concentration of snow and graupel increase significantly? Due to the aerosol invigoration effect? You need to explain. 8. Line 317-321, 351-353: Why do the aerosol impacts on stronger and weaker convective systems quite different? You should explain briefly here since the discussions in Sections 3.3 and 3.4 are far away. I think your finding that fire aerosols tend to invigorate weak convection but suppress deep convection is generally consistent with and could be better supported by previous observation-based studies (e.g., Jiang et al., 2018, Nature Communications, DOI 10.1038/s41467-018-06280-4; Zhao et al., 2018, GRL, DOI 10.1002/2018GL077261), which showed that smoke aerosols generally suppress deep convection and convection-generated ice clouds. 9. Line 381-408: This part is difficult to follow and should be better organized. The author intends to investigate the dependency of the aerosol impact on convective strength (Line 381-382). This question is discussed for r1, but not clearly for r2. From the current text, I am

not sure how the aerosol impacts differ for convective systems with different strength in r2. The same problem exists in the conclusion section. Also, why do you introduce daily maximum and minimum rainfall? A few transitional sentences are needed. Line 391-392: Better to mention clearly that this refers to r1. 10. Figures 5, 6: Some texts in the figures are too small to be visible.

---

## Author Comment (AC1) · 15 Nov 2019

**Responses to the Comments of the Anonymous Referee #1**

We very much appreciate the constructive comments and suggestions from this reviewer. Our point-by-point responses to the reviewer's comments are as follows (the reviewer's comments are marked in Italic font).

*Comments:*

*This study investigates the impact of biomass burning aerosols on convective systems in the Sumatra and Borneo regions of Southeast Asia using the WRF-Chem model. Considering the large uncertainty in the interactions between aerosols, particularly those from biomass burning, and convective clouds, this study advances our understanding of the complicated and competing physical processes that governs the net effect of biomass-burning aerosols. The manuscript is generally well written. I think it can be considered for publication after the author addresses the following comments and suggestions.*

*1. Abstract: The descriptions after Line 45 are much too general. The author mentioned several times that fire aerosols have "significant/substantial impacts" on convection. What exactly are these impacts? I believe the author should summarize their main findings here so that the abstract can be more informative.*

We thank the reviewer's suggestion. We have modified the abstract as: "Results from selected cases of convective events have shown significant impacts of fire aerosols specifically on the weak convections by increasing the quantities of hydrometeors and rainfall in both Sumatra and Borneo regions. Statistical analysis over the fire season also suggests that fire aerosols have impacts on the nocturnal convections associated with the local anticyclonic circulation in the western Borneo and then weakened the nocturnal rainfall intensity by about 9%. Such an effect is likely come from the near surface heating by absorbing aerosols emitted from fires that could weaken land breezes and thus the convergence of anticyclonic circulation."

*2. Line 173-175: How did you treat emissions from the flaming vs smoldering phases when calculating plume rise? A previous study (Shi et al., 2019, JGR-Atmospheres, DOI 10.1029/2019JD030472) has shown that the fraction of smoldering-phase emissions has a large impact on plume rise and fire-induced aerosol concentrations.*

The current plume rise algorithm in WRF-Chem is based on the burning vegetation types not burning phases. In the reality, most peatland fires burn in smoldering-phase and most fire aerosols concentrate near surface. Shi et al. (2019) pointed out that not considering the characteristics of smoldering phase of burning in the model could lead to underestimated fire emissions and thus near surface fire aerosol concentration.

Our study has considered the first issue (the vegetation types) and we have made corresponding modification to WRF-Chem plume model. As mentioned in the manuscript, for peatland fire, we have set its heat flux as 4.4 kW m$^{-2}$, which is the same as that of savanna burning while differs significantly from that of the tropical forest burning in 30 kW m$^{-2}$. Furthermore, we have limited the plume injection height of peat fire by a ceiling of 700 m above

the ground based on remote sensing retrieval from Tosca et al. (2011). The injection height for tropical peat fire in our modeling was thus derived based on this new algorithm. We agree with the reviewer that the phase of burning should be considered more carefully in future efforts in deriving fire emission inventory. We have added a sentence in Lines 187-190 of the manuscript as: "Note that the current fire emission inventories could underestimate near surface fire aerosol concentration by ignoring some of the characteristics of smoldering burning as well (Shi et al., 2019)."

*3. Line 185-186: I think it's not accurate to use "fossil fuel emissions" here. "Anthropogenic emissions" may be a better term. Many anthropogenic emissions do not originate from fossil fuels, such as VOC emissions from solvent use, NH3 emissions from agricultural activities, and emissions from household biomass fuels.*

We have modified the sentence to "Two numerical simulations, both included anthropogenic emissions (mainly fossil fuel emissions) while either with and without the biomass burning emissions (labeled as FFBB and FF, respectively) …"

*4. Line 191-193: This is an important point. You may want to show the data in SI.*

We have added Fig. S1, the time series of domain-averaged monthly mean $PM_{2.5}$ emissions from FINN and precipitation rate from TRMM dataset, in the supplement.

*5. Section 3.1.2: Since fire emissions have a large day-to-day variability, I think the monthly mean AOD may not be suitable for evaluating the model performance. I suggest to use daily product (MOD08_D3) instead. Also, the author argues that the higher simulated AOD than observations is because "a high spatiotemporal resolution in our simulation enables the model to capture episodic fire events better". I think comparing with daily AOD observations could help to confirm whether this argument is true or not.*

We appreciate and actually agree on the reviewer's point that fire emissions have a large day-to-day variability. However, due to the frequent appearance of convective systems, MODIS AOD data are often derived based on limited non-cloudy pixels in this region. In this sense, we believe that the comparison of MODIS AOD in a longer period (here we select for monthly) might better serve the purpose. In our pervious study, we have performed more quantitative comparisons of fire pollutants between modeled results and ground-based observations. In Lee et al. (2018), the comparisons of daily $PM_{10}$, CO, $O_3$ and visibility have demonstrated that the model is capable to capture episodic fire events during a long-term simulation.

*6. Line 303: but smaller number?*

The process of cloud droplet collection by rain increases the mass of rain while causes no change to the number of raindrops. We have modified the sentence to "Larger raindrops combining with smaller cloud droplets in FFBB can enhance the efficiency of cloud droplet collection by rain and thus increase rain water mass but cause no change to the number of raindrops, possibly compensating the decrease of rain water mass resulted from a lowered autoconversion."

*7. Line 307-308: Why do the mass concentration of snow and graupel increase significantly? Due to the aerosol invigoration effect? You need to explain.*

We have added following sentences to explain the change of snow and graupel mass concentration in Lines 335-343 of the revised manuscript:
"Our result is consistent with that of Lin et al. (2006), which suggested that biomass burning aerosols could invigorate convection and then increase precipitation based on satellite observations. The aerosol invigoration effect is referred to such a hypothetic process that increasing number of smaller cloud droplets due to higher aerosol concentration would reduce the efficiency of raindrop formation from self-collection among cloud droplets, and thus further slowdown the loss of these small droplets from being collected by larger raindrops and allow more of them reach high altitudes, where they would eventually collected by ice particles through riming, causing release of latent heat to enhance updraft."

*8. Line 317-321, 351-353: Why do the aerosol impacts on stronger and weaker convective systems quite different? You should explain briefly here since the discussions in Sections 3.3 and 3.4 are far away. I think your finding that fire aerosols tend to invigorate weak convection but suppress deep convection is generally consistent with and could be better supported by previous observation-based studies (e.g., Jiang et al., 2018, Nature Communications, DOI 10.1038/s41467-018-06280-4; Zhao et al., 2018, GRL, DOI 10.1002/2018GL077261), which showed that smoke aerosols generally suppress deep convection and convection-generated ice clouds.*

We thank the reviewer's suggestion. We have added the following sentences in Lines 396-409 of the revised manuscript:
"Our results show that fire aerosols tend to invigorate weak convection but suppress deep convection in both Sumatra region (r1) and Borneo region (r2). As mentioned before, increasing the number of smaller cloud droplets due to higher aerosol concentration resulted from fire would reduce the efficiency of raindrop formation through the warm-rain processes, thus allowing more cloud droplets reach high altitudes to be eventually collected by ice particles through riming, causing release of latent heat to invigorate updraft while enhancing precipitation through melting of fallen ice particles (Wang, 2005). These processes appear to be more effective to weak convections than deep convections and were in fact well-simulated in the former cases. The results are also consistent with some previous observation-based studies (Jiang et al., 2018; Zhao et al., 2018). Jiang et al. (2018) and Zhao et al. (2018) both concluded that an increase of fire aerosols generally reduces cloud optical thickness of deep convection while Zhao et al. (2018) further showed that fire aerosols tend to invigorate weak convection for small-to-moderate aerosol loadings."

*9. Line 381-408: This part is difficult to follow and should be better organized. The author intends to investigate the dependency of the aerosol impact on convective strength (Line 381-382). This question is discussed for r1, but not clearly for r2. From the current text, I am not sure how the aerosol impacts differ for convective systems with different strength in r2. The same problem exists in the conclusion section. Also, why do you introduce daily maximum and*

*minimum rainfall? A few transitional sentences are needed. Line 391-392: Better to mention clearly that this refers to r1.*

We have made an effort to clarify the commented discussions in Lines 431-448 of the revised manuscript:

"In Sect. 3.2, we have discussed the significant rainfall increase occurred in the weak convective systems after adding fire aerosols due to aerosol invigoration effect.  On one hand, regardless the strength of convection, the mean 3-hourly rainfall during the fire periods is 1.06±0.85 mm in FF and 1.09±0.86 mm in FFBB over the Sumatra region (r1), and statistically it does not change significantly in responding to fire aerosols.  The rainfall difference in the Borneo region (r2) between FF and FFBB is also insignificant (1.32±1.20 mm 3hrs$^{-1}$ in FF versus 1.35±1.14 mm 3hrs$^{-1}$ in FFBB).  On the other hand, we have found that the impacts of fire aerosols appear in several other rainfall patterns.  For instance, the daily maximum and minimum rainfalls display clear differences between the FFBB and FF simulations, specifically in r2 rather than in r1 (Fig. 9).  While for r1, the impacts of fire aerosol are reflected in event-wise statistics, e.g., higher event-wise maximum and minimum rainfall intensity in FFBB than in FF, identified in 30 out of 54 convective events in total.  These are mostly weak convective events in r1.  Interestingly, somewhat opposite to the rainfall statistics in r1, the intensity of event-wise maximum and minimum rainfall in r2 is higher in FF than in FFBB.  The daily rainfall peak of 3-hr rainfall in r1 is mostly less than 3 mm; in comparison, one-third of convective events in r2 have daily maximum 3-hr rainfall exceeding 3 mm (Fig. 9c), suggesting that the convective systems in r2 tend to develop stronger than in r1 and the fire aerosols significantly suppress the maximum rainfall intensity of strong convections in r1.  …"

*10. Figures 5, 6: Some texts in the figures are too small to be visible.*

Texts in the figures have been modified in the revised manuscript.

Reference:

Jiang, J.H. *et al.*, Contrasting effects on deep convective clouds by different types of aerosols, *Nature Communications* **9**(2018), p. 3874.

Lee, H.-H. *et al.*, Impacts of air pollutants from fire and non-fire emissions on the regional air quality in Southeast Asia, *Atmos. Chem. Phys.* **18**(2018), pp. 6141-6156.

Lin, J.C., Matsui, T., Pielke Sr., R.A., Kummerow, C., Effects of biomass-burning-derived aerosols on precipitation and clouds in the Amazon Basin: a satellite-based empirical study, *Journal of Geophysical Research: Atmospheres* **111**(2006).

Shi, H. *et al.*, Modeling Study of the Air Quality Impact of Record-Breaking Southern California Wildfires in December 2017, *Journal of Geophysical Research: Atmospheres* **124**(2019), pp. 6554-6570.

Wang, C., A modeling study of the response of tropical deep convection to the increase of cloud condensation nuclei concentration: 1. Dynamics and microphysics, *Journal of Geophysical Research: Atmospheres* **110**(2005), p. D21211.

Zhao, B. *et al.*, Type-Dependent Responses of Ice Cloud Properties to Aerosols From Satellite Retrievals, *Geophysical Research Letters* **45**(2018), pp. 3297-3306.

---

## Author Comment (AC2) · 15 Nov 2019

**Responses to the Comments of the Anonymous Referee #2**

We very much appreciate the constructive comments and suggestions from this reviewer. Our point-by-point responses to the reviewer's comments are as follows (the reviewer's comments are marked in Italic font).

***Comments:***

*This paper examines the impacts of biomass burning aerosols on convective systems over the northern Sumatra and the western Borneo in the Maritime Continent based on based on long-term WRF-Chem simulations. While the paper is well written and interesting, there are some concerns that need to be addressed before the paper being publishable.*

*(1) The resolution of inner domain at 5 km is still too coarse for simulating convective clouds.*

We agree with the reviewer that 5 km is still not an ideally fine resolution for simulating convective clouds, although commonly previous studies have shown that WRF model with a similar resolution can still reflect many critical characteristics of deep convection without using convection parameterization (e.g., Wagner et al., 2018). Specifically, because of the purpose of this study and the availability of computational resource, we have decided to use a 5 km resolution with cumulus scheme off. Based on our model evaluation, especially the comparison of sounding profiles, the model under the current configuration can capture the major characters of the convective systems very well.

In the revised manuscript, we have added following sentences in Section 2.1, Lines 162-170:

"Owing to the main purpose of this study to reveal fire aerosol-convection interaction through modeling a large quantity of convective systems continually over a relatively long period, and the computational resource available to us as well, we have adopted a 5 km horizontal resolution which excluding cumulus parameterization scheme. Previous studies have shown that WRF model with a similar resolution without convection parameterization can still capture many critical characteristics of deep convection (Wagner et al., 2018). Our model evaluation, especially through the comparison of modeled results with sounding profiles, has demonstrated the same."

*(2) The section of selected cases analysis looks vague and needs more detailed analysis. The impacts of aerosol on precipitation are very complicated (different mechanisms on different clouds under different conditions). It is too bold to get the conclusion just based on three cases even with cloud types unknown. There are so many convective cases that could be categorized them and analyzed in detail.*

We have made our best effort to clarify the related discussions in the revised manuscript (see also our response to the Reviewer #1). First of all, we have made it clear that our simulations and analyses cover all the different cases, though we have chosen to identify different aspects of the impacts of fire aerosols through a different sets of analyses, ranging from deriving case-wise statistics to performing seasonal analysis.

The analyses based on selected three cases from each of the two study regions are just one of these. To avoid the impression that our conclusions were drawn from only three cases, we have made several revisions in the manuscript, One of the revised discussions in Lines 429-448 are: "In Sect. 3.2, we have discussed the significant rainfall increase occurred in the weak convective systems after adding fire aerosols due to aerosol invigoration effect. On one hand, regardless the strength of convection, the mean 3-hourly rainfall during the fire periods is 1.06±0.85 mm in FF and 1.09±0.86 mm in FFBB over the Sumatra region (r1), and statistically it does not change significantly in responding to fire aerosols. The rainfall difference in the Borneo region (r2) between FF and FFBB is also insignificant (1.32±1.20 mm 3hrs$^{-1}$ in FF versus 1.35±1.14 mm 3hrs$^{-1}$ in FFBB). On the other hand, we have found that the impacts of fire aerosols appear in several other rainfall patterns. For instance, the daily maximum and minimum rainfalls display clear differences between the FFBB and FF simulations, specifically in r2 rather than in r1 (Fig. 9). While for r1, the impacts of fire aerosol are reflected in event-wise statistics, e.g., higher event-wise maximum and minimum rainfall intensity in FFBB than in FF, identified in 30 out of 54 convective events in total. These are mostly weak convective events in r1. Interestingly, somewhat opposite to the rainfall statistics in r1, the intensity of event-wise maximum and minimum rainfall in r2 is higher in FF than in FFBB. The daily rainfall peak of 3-hr rainfall in r1 is mostly less than 3 mm; in comparison, one-third of convective events in r2 have daily maximum 3-hr rainfall exceeding 3 mm (Fig. 9c), suggesting that the convective systems in r2 tend to develop stronger than in r1 and the fire aerosols significantly suppress the maximum rainfall intensity of strong convections in r1. …"

We have specifically enhanced our discussion regarding mechanisms of fire aerosol-convection impacts. On example is the following added discussions in Line 333-341: "Our result is consistent with that of Lin et al. (2006), which suggested that biomass burning aerosols could invigorate convection and then increase precipitation based on satellite observations. The aerosol invigoration effect is referred to such a hypothetic process that increasing number of smaller cloud droplets due to higher aerosol concentration would reduce the efficiency of raindrop formation from self-collection among cloud droplets, and thus further slowdown the loss of these small droplets from being collected by larger raindrops and allow more of them reach high altitudes, where they would eventually collected by ice particles through riming, causing release of latent heat to enhance updraft."

Regarding defining a weak or strong convective system in the case study, 3 mm 3hr$^{-1}$ of the averaged rainfall could be used as a threshold. We have clarified this in the revised manuscript.

*(3) The heating effect of fire aerosols seems too weak to have significant influence on circulation.*

The temperature increase from aerosol absorption is not necessarily too weak because our analysis did identify clear change of vertical velocity owing to the aerosol heating effect. This seems also consistent with the analysis of Zhang et al. (2019). Indeed, should the heat flux generated by fires be incorporated in the model, the warming effects from biomass burning would be much stronger and also persist in nocturnal timeframe.

We have added sentences in Lines 522-527 of the revised manuscript as: "Based on our analysis, the temperature increase is mainly associated with the thermodynamic perturbation from the absorption of sunlight by fire aerosols. This seems also consistent with the analysis of Zhang et al. (2019). Indeed, should the heat flux generated by fires be incorporated in the model,

the warming effects from biomass burning would be much stronger and also persist in nocturnal timeframe."

*Here are also some specific comments:*
*1. Line 153: What is the time frequency of nudging?*

The time frequency of nudging is every 6 hours. We have added this information in the revised manuscript.

*2. Line 183: needless parenthesis*

Modified.

*3. Line 205: How and why were these convective systems selected? Why only three?*

The selected cases in Section 3.2 are chosen randomly from different fire periods of the two study regions. We did not set any criteria initially when we chose these cases. All these points have been clarified in the revised manuscript (Lines 221-224):
"The selected cases are chosen randomly from the different fire periods of the two study regions. We did not set any criteria initially when we chose these cases. After we analyzed all cases, 3 mm 3hr$^{-1}$ was set as the threshold to distinguish weak and strong convections."
These cases were used to discuss modeled characteristics of individual cases and make comparison between cases from different regions without considering their weights in the overall case-wise statistics. We have actually made effort to avoid leaving any impression about whether they are representative in their corresponding case population. The ensemble characteristics of each case population are defined by their case-wise statistics.

*4. Line 233: Were the model results interpolated to the resolution of TRMM before doing the comparison?*

The modeled rainfall in FFBB has been interpolated to the resolution of TRMM for the comparison in Figure 3. Figure 2a is also made by following this procedure. However, we have just realized that the original Fig. 2b was not made after modeled data being remapped into TRMM grids; therefore, we have reprocessed data and replotted Fig. 2b in the revised manuscript to be consistent with other figures. We appreciate the reviewer for pointing out this.

*5. Line 260: Why only this example sounding is shown? You may compare with many other cases and even show a statistical comparison.*

We choose this example sounding in the main text because we have cloud vertical structure from CALIPSO for the same case. We have now added the sounding comparison of other 5 cases in the supplement.

*6. Line 275: Only one case captured by CALIPSO?*

We have compared more than 50 modeled convections during the fire season and within the simulation domains. Specifically, for the six selected cases in case study, only one case was captured by CALIPSO. The others captured by CALIPSO are not among the cases in the case study (some are even out of our analyzed domains). This is the reason why we only discussed this case in our case study discussion. We have mentioned this point in Lines 302-304 of the revised manuscript.

*7. Line 311-314: It is confusing. Aerosol impact on ice-phase microphysical processes is still considered in Morrison through the CCN effect. It is the IN effect of aerosol that is missed.*

We thank the reviewer for indicating this. We have added "In our model configuration, fire aerosol can still affect ice process, however, through CCN effect rather than serving directly as ice nuclei." into Lines 344-346 in the revised manuscript.

*8. Line 315-321: More background information of these cases are necessary. You just simply saying that one case has weaker convective systems than other two. This is too ambiguous.*

The reviewer's comment is well taken. We have added the sounding profiles of all six cases in the supplement to present the environmental condition of each of these convections. We have also added a sentence of: "After we analyzed all cases, 3 mm 3hr$^{-1}$ was set as the threshold to distinguish weak and strong convections" into Lines 223-224 in the revised manuscript.

*9. Line 454: The temperature increase seems too small. Is this significant? Maybe the difference is within the model simulating error range.*

As we replied in the general comment (3), based on our analysis we believe this temperature increase is mainly associated with the thermodynamic perturbation from the absorption of sunlight by fire aerosols. It is also consistent with the analysis of Zhang et al. (2019). Again, should the heat flux generated by fires be incorporated in the model, the warming effects from biomass burning would be much stronger and also persist in nocturnal timeframe.

*10. Line 455-457: No figure showing this conclusion. How much land breeze and surface convergence is weakened.*

We have added Fig. 11 to demonstrate this conclusion in the revised manuscript.

Reference:

Lin, J.C., Matsui, T., Pielke Sr., R.A., Kummerow, C., Effects of biomass-burning-derived aerosols on precipitation and clouds in the Amazon Basin: a satellite-based empirical study, *Journal of Geophysical Research: Atmospheres* **111**(2006).
Wagner, A., Heinzeller, D., Wagner, S., Rummler, T., Kunstmann, H., Explicit Convection and Scale-Aware Cumulus Parameterizations: High-Resolution Simulations over Areas of Different Topography in Germany, *Monthly Weather Review* **146**(2018), pp. 1925-1944.

Zhang, Y., Fan, J., Logan, T., Li, Z., Homeyer, C.R., Wildfire impact on environmental thermodynamics and severe convective storms, *Geophysical Research Letters* **0**(2019).

---

## Author Comment (AC3) · 15 Nov 2019

The comment was uploaded in the form of a supplement: https://www.atmos-chem-phys-discuss.net/acp-2019-632/acp-2019-632-AC3-supplement.pdf

---

## Author Comment (AC4) · 15 Nov 2019

**Responses to the Comments of the Anonymous Referee #3**

We very much appreciate the constructive comments and suggestions from this reviewer. Our point-by-point responses to the reviewer's comments are as follows (the reviewer's comments are marked in Italic font).

*Comments:*

*The authors attempt to investigate the fire aerosol-cloud-precipitation interactions by conducting modeling sensitivity studies. The performance of WRF-CHEM simulations were fully evaluated, and the responses of cloud microphysics and precipitation amount to fire aerosols were carefully quantified. However, I still have some minor issues about this work prior to its publication.*

*1. In the discussions in sections 3.1 and 3.2, it appears that the responses of cloud microphysics properties and precipitation to fire aerosols are sensitive to convection intensity of the systems selected for case studies, but the authors didn't show what are the criteria to determine the systems are convective weak. At least, some basic description about the selected convective systems should be provided so that the readership could have some sense about the convection strength of each system.*

We thank the reviewer for pointing out this. The selected cases are chosen randomly from the difference fire periods of the two study regions. We did not set any criteria initially when we chose these cases. After we analyzed all cases, 3 mm 3hr$^{-1}$ was set as the threshold to distinguish weak and strong convections. We have clarified this in Lines 223-224 of the revised manuscript.

*2. Related to point 1, is that possible to do statistical analysis of fire periods for weak and strong convective systems separately? Since the weak systems are more sensitive to fire aerosols, I would expect that there might be more significant differences in cloud properties or precipitation between fire aerosol case and non-fire aerosol case when looking at those weak systems.*

The reviewer's suggestion is well taken. We have thus analyzed the weak against strong convections. Practically, however, it is difficult to directly use the same statistics in the case study to all the convections in different types, not mentioning the relativeness of the criterion for separating them when put all the cases together. Instead, we have performed statistics based on domain averages. Here again, the threshold of 3 mm 3hr$^{-1}$ for several selected cases is hard to apply to the domain wise statistical analysis, this is because that the domain-averaged rainfall in the statistical analysis is generally weaker than the averaged rainfall in the case study.

Therefore, we choose to use 1.25 mm 3hr$^{-1}$ of the domain-averaged rainfall to separate weak from strong convective systems. We find that the conclusions regarding differences of hydrometers and rainfall in the weak systems between the FF and FFBB experiments stay the same, and such differences are still not significant. We have added one paragraph in Sect. 3.3 for the statistical analysis for weak and strong convective systems during fire periods. In addition, Table S1 has been added to show average daily-rainfall of FFBB and FF for strong and weak convections during fire periods over the Sumatra region (r1) and Borneo region (r2), and Fig. S8 to indicate the mean hydrometeor differences in percentage between FFBB and FF.

*3. Are the fire periods shown in Table 2 the time periods during which the fire aerosols are continuously emitted into atmosphere? I just want to make sure that the cloud systems selected for statistics of fire season are those which were indeed influenced by fire aerosols. That means the selected cloud systems for analysis concurrently occurred with fire events.*

In our FFBB simulation, the fire aerosols were continuously emitted into the atmosphere. We present the time series of fire counts in the two study regions in Fig. S2.

*4. Please add uncertainties of precipitation for each case in Figure 9.*

We assume the figure referred by the reviewer was Fig. 10. The uncertainties have been added in Fig. 10 in the revised version of manuscript.

*5. In section 3.4, the impacts of fire aerosols on local circulations like land/sea breeze are not evident. Some figures like the mean wind fields for fire aerosol and non-fire aerosol cases to show their difference would be helpful.*

We thank the reviewer's comment. We have added a new figure (Fig. 11) in the revised manuscript to illustrate the sea breeze increase in FFBB during the daytime (20 LTC) and the land breeze decrease in FFBB during the night daytime (2 LTC).

---

## Author Comment (AC5) · 15 Nov 2019

The comment was uploaded in the form of a supplement: https://www.atmos-chem-phys-discuss.net/acp-2019-632/acp-2019-632-AC5-supplement.pdf

---

## Author Response (AR2)

**Responses to the Comments of the Anonymous Referee #2**

We appreciate the further comments and suggestions from this reviewer. Our point-by-point responses to the reviewer's comments are as follows (the reviewer's comments are marked in Italic font).

*Comments:*

*This paper examines the impacts of biomass burning aerosols on convective systems over the northern Sumatra and the western Borneo in the Maritime Continent based on based on longterm WRF-Chem simulations. The authors have made efforts to address reviewers' comments and improve the manuscript. The improvements make the paper much clearer and its conclusions better defended. I have a few suggestions and questions outlined below, which I believe can be addressed with minor revisions prior to publication.*

*1. Line 221-224: Why not define the threshold first and then select the representative case for analysis?*

As we mentioned in the manuscript, the selected cases are chosen randomly from different fire periods of the two study regions. We did not know what should be a proper criterion to distinguish weak from strong convections among these samples beforehand. It was actually not our initial intention as well because the selected samples might not represent the overall statistics – notice that the weak versus strong is rather relative. After we analyzed all cases, we found that biomass burning aerosols impact on relative weak and strong convections differently. Thus, we set 3 mm 3hr$^{-1}$ as the threshold to distinguish weak and strong convections for the following discussion in the manuscript.

*2. Section 3.2: The precipitation and hydrometeor properties are analyzed. How is the convective intensity? e.g., updrafts.*

Indeed, discussions in Sect. 3.2 are about the impacts of biomass burning aerosols on microphysical features and precipitation. The discussion of convective intensity is mainly in Sect. 3.4.

*3. Line 335-341: This is only the cold-phase invigoration effect (Rosenfeld 2008). For tropical regions with high humidity, small size aerosols may also show warm-phase invigoration due to additional activation (Fan et al, 2018).*

We thank the reviewer's suggestion. We added the Rosenfeld et al. (2008) for the cold-phase invigoration effect and "For tropical regions with high humidity, additional aerosols may also lead to the warm-phase invigoration due to the consequent enhancement in total condensed water quantity and thus latent heat release (Wang, 2005; Fan et al., 2018)" in the revised manuscript.

*4.Line 480: biomass burning activities may include the released heat, aerosols and water*

*vapor. Since you only discuss the effect of biomass burning aerosol, I would suggest to change "activities" to "aerosols".*

Modified as the reviewer suggested.

*5. Fig 11: What are 20LST and 02LST showing? The peak of difference? It is better to show a time series of sea breeze and land breeze change and link with the near surface temperature change.*

We appreciate the comment of the reviewer. We have added the following sentences in the manuscript: "Figure 11 illustrates the sea breeze increase in FFBB during the daytime (20 LST) and the land breeze decrease in FFBB during the nighttime (2 LST). Twenty LST and 2 LST are chosen here because of the peak of difference".

The time series is hard to see the transition of sea breeze and land breeze change so that we present the peak of difference in the figure. Adding surface temperature change may make the figure too busy to read. For time series change of temperature and other parameters, the readers can get the information from Fig. 12.

[revised manuscript text omitted]